# Isolated Lumbar Extension Resistance Exercise in Limited Range of Motion for Patients with Lumbar Radiculopathy and Disk Herniation—Clinical Outcome and Influencing Factors

**DOI:** 10.3390/jcm10112430

**Published:** 2021-05-30

**Authors:** Witold Golonka, Christoph Raschka, Vahid M. Harandi, Bruno Domokos, Håkan Alfredson, Florian Maria Alfen, Christoph Spang

**Affiliations:** 1Department of Sports Science, University of Würzburg, 97082 Würzburg, Germany; witold.golonka@spine-research.de (W.G.); christoph.raschka@uni-wuerzburg.de (C.R.); bruno.domokos@spine-research.de (B.D.); 2Private Orthopedic Spine Center, 97080 Würzburg, Germany; praxis@dr-alfen.de; 3Department of Experimental Medical Science, Lund University, 221 84 Lund, Sweden; vahid.m_harandi@med.lu.se; 4Department of Community Research and Rehabilitation, Umeå University, 901 87 Umeå, Sweden; hakan.alfredson@umu.se; 5Institute of Sport, Exercise and Health (ISEH), London W1T 7HA, UK

**Keywords:** disk herniation, radiculopathy, low back pain, ILEX, exercise, conservative treatment

## Abstract

(1) Background: Reconditioning of the paraspinal lumbar extensor muscles by isolated lumbar extension resistance exercises (ILEX) has shown good clinical results for patients with chronic unspecific low back pain. However, the clinical value and safety for patients with specific spine pathologies is unclear. In this study, clinical outcome and influencing factors were retrospectively analyzed for patients with lumbar disk herniation (LDH) and radiculopathy. (2) Methods: 189 consecutive patients (123 men and 66 women; mean age, 36 years) with clinically diagnosed LDH and relative indications for surgery started a 9-week rehabilitation program (2x/week) including ILEX in limited range of motion (ROM) adjusted to patients’ symptoms. Patients diagnosed with advanced levels of spine degeneration were excluded. Pain/radiculopathy (PR), influence on mental health (IOMH), satisfaction rates were measured via Numeric Rating Scales (NRS, 0–10), and overall clinical outcome was stated in % (100% = full recovery). Isometric extension strength was tested before and after the program. (3) Results: 168 patients (88.9%) completed the program. For 162 out of 168 patients (96.4%) there was a significant reduction of clinical symptoms, whereas 6 patients reported no changes in symptoms. Scores (mean) for symptom intensity decreased from 4.2 (±1.5) to 1.9 (±1.5) (*p* < 0.001), the impact on mental health decreased from 5.9 (±2.3) to 2.4 (±2.0) (*p* < 0.001). There was a (weak) correlation between lower scores for PR and IOMH before the study and better clinical outcomes; PR also weakly correlated with satisfaction. Other factors such as age, strength increase, level/location and number of LDH did not have a significant impact on the clinical results. (4) Conclusion: The results indicate that ILEX in limited ROM can be an effective treatment for the majority of patients with LDH. For patients with high pain levels, the results are less consistent, and surgery may be considered.

## 1. Introduction

Lumbosacral radiculopathy is a frequent clinical condition that results from compression of one or more spinal nerve roots. It is mainly characterized by radiating leg pain and paresthesia, as well as clinical signs of neurological impairment [1]. The prevalence ranges in studies from 1.2% to 43.3% [2]. Painful radiculopathy is an entity of neuropathic pain [3]. In most patients, it is caused by lumbar disk herniations (LDH)—localized displacements of disk material beyond the margins of the intervertebral disk space accompanied by acute vasodilation and migration of inflammatory cells [1,4]. A sufficient diagnosis for LDH is best made with imaging tools such as magnetic resonance imaging (MRI), showing high reliability [5]. In contrast, physical and neurological examinations have only limited overall diagnostic accuracy in detecting LDH [6,7]. 

Although immediate surgical intervention may be necessary for patients with permanent lower limb weakness, paralysis or impairment of bowel and bladder control, conservative treatment should be preferred as the initial management method for the majority of patients with LDH [8]. Studies comparing surgical and conservative methods have shown that surgery may result in a rapid relief of symptoms, but after 2–5 years there were no significant differences concerning clinical outcome [9,10]. In a meta-analysis by Zhong and colleagues, it was shown that the average incidence of spontaneous regression and reabsorption of LDH is quite high (67%) [11]. As there were huge differences between studies and their corresponding countries, the authors hypothesized that national medical standards and the type of conservative approach significantly influence the rate of LDH regression and consequently long-term rehabilitation [11]. So far, systematic reviews and clinical trials have failed to identify a single conservative treatment that has proven to be successful, safe and superior to other treatments for patients with LDH [8]. This could be partly explained by the fact that patients were often recruited based on clinical rather than radiological examination. According to Hahne and colleagues, it is plausible that this drawback may have diluted the treatment effects, as patients with other diagnoses such as stenosis and spondylolisthesis were included in those studies as well [8]. Thus, studies on new conservative treatments and properly diagnosed homogenous patient groups are needed.

Multiple studies have shown that chronic pain conditions in the lower back region are associated with a deconditioning process in the paraspinal back extensor muscles [12]. According to several experts, this process starts with early-stage muscle inhibition as a response to acute pain and develops to pronounced muscle atrophy and local fat infiltration in chronic stages [13]. Therapeutic approaches aiming to restore these local extensor muscles, such as isolated lumbar extension resistance exercise (ILEX) with an exercise device that offers appropriate pelvic restraint, have resulted in good clinical outcomes for patients with chronic unspecific low back pain [14,15]. In patients with LDH, there is a similar pattern of local deconditioning. In a study by Fortin and colleagues, significantly greater fat infiltration on the side with pain and radiculopathy symptoms, and at spinal levels adjacent to the lumbar disk herniation, was found [16]. Based on this correlation between the occurrence of LDH and local paraspinal muscle deconditioning, it seems logical to examine the safety and clinical outcome of the ILEX rehabilitation program in patients with diagnosed LDH and radiculopathy. So far, ILEX has only been studied as a tool for postoperative rehabilitation [17] but not as a treatment option for this patient group. Studies on exercise interventions for LDH have mainly focused on therapeutic core stabilization, often using body weight exercises, or on post-operative rehabilitation [18,19,20,21,22]. However, based on studies from Steele and colleagues, ILEX is superior for strengthening and reconditioning of paraspinal lumbar extensor muscles [12,15] and ensures safe isolated heavy loading exercises in controlled movement (limited range of motion) for the patient [13]. 

To the best of our knowledge, there is no study that has so far evaluated the clinical outcome of ILEX in a homogenous patient group with clinically and MRI-diagnosed lumbar disk herniation together with lower limb radiculopathy and sensory impairment. In this current study, we analyzed the clinical outcomes of a consecutive patient series with LDH but without additional spinal disorders or age-related degeneration. Furthermore, we evaluated the effect of influencing factors such as pain duration, severity of symptoms, age, disk level and strength increase on clinical outcome.

## 2. Materials and Methods

### 2.1. Patients

Patients included in this study were part of a large consecutive case series. All participated in a medical strengthening therapy program (18 sessions in total) including one set of ILEX exercise with limited ROM in the same spine center (2002–2019; Dr. Florian Alfen, Würzburg, Germany). Via retrospective data analysis, clinical outcome measures were analyzed for patients fulfilling the following criteria: patients who started the program and had one or more diagnosed lumbar disk herniations on an MRI that was not older than 3 months prior to the start of the exercise program. Patient data were not included when the patient had undergone previous surgical procedures in the spine region. Patients were also not considered when they performed less than 3 sessions or dropped out due to non-medical reasons. In order to exclude the impact of spine degeneration as a potential additional source of symptoms, those with diagnosed degenerative and inflammatory disorders (e.g., facet joint arthritis, stenosis, spondylolisthesis) were excluded as well. Assuming substantial degenerative changes in patients older than 50 years of age, those patients were excluded as well.

Altogether, 189 patients (123 men and 66 women; mean age, 36 years) fulfilled the above inclusion criteria, and outcome data from the below-described ILEX exercise rehabilitation program were analyzed. There were 21 patients who did not finish the full treatment of 18 sessions in the suggested time frame due to medical reasons (average drop out after 12.0 sessions): persistent symptoms and immediate spine surgery suggested by the local medical consultant (FA) (*n* = 14); persistent symptoms and not willing to continue despite no surgical indication by the medial consultant (*n* = 7). 

In total, clinical data on clinical outcomes from 168 patients (110 men, 58 women with a mean age of 37 years (range 16–50) were further analyzed in detail. All the included patients had tried different conservative treatment regimens previously with only scarce success. The majority (*n* = 140, 84%) had performed different types of physiotherapy (manual therapy, physical therapy, exercise). Most patients (*n* = 134, 80%) were physically active in different sport activities such as endurance sports (*n* = 100, 59%), fitness and strength training (*n* = 41, 24%) or team sports (*n* = 46, 27%).

### 2.2. Clinical Examination

All patients underwent clinical examination by an experienced orthopedic consultant and spine surgeon (FA) in the same local spine center. Medical history was reviewed, and MRI images were assessed. Careful neurological examination including sensory, motor and standard reflex testing of the lower limbs was performed [5]. When there were signs of permanent lower limb weakness, paralysis or impairment of bowel and bladder control, patients were recommended to undergo immediate surgery, thus not being part of this study (see exclusion criteria). Patients that had acceptable motor function, but signs of sensory impairment and high pain intensity were assigned for conservative rehabilitation and recruited for the current study. Clinical examinations were generally repeated after 6, 12 and 18 exercise sessions. 

Almost all patients in this study (*n* = 159, 95%) suffered from radiculopathy in different regions: pelvis/hip (*n* = 51, 30%), legs (*n* = 133, 79%) and buttocks (*n* = 99, 59%) (Figure 1a). LDH mostly occurred at L5/S1 (*n* = 131, 78%), followed by L4/L5 (*n* = 71, 42%), L3/L4 (*n* = 14, 8%), L2/L3 (*n* = 1, <1%) and L1/L2 (*n* = 1, <1%) (Figure 1b). Multiple disk herniations were detected in 40 patients (24%). In addition, 50 patients (31%) were on painkillers and related medication at the start of the therapy program. In total, 80 patients had taken medication in the past against their symptoms. Apart from one patient with high blood pressure, one with gastritis and one with cirrhosis of the kidney, there were no chronic internal and inflammatory diseases among the patients. 

### 2.3. Lumbar Extension Machine

According to previous studies, isolated training of paraspinal lumbar extensor muscles requires pelvis stabilization in order to increase local muscle loading but also to eliminate the contribution of the gluteal and hamstring muscles that cause backward rotation of the pelvis [23,24]. The restraint system used is based on a previously described lumbar extension machine system (Figure 2) and has been used in multiple studies on patients with unspecific low back pain [12,14]. After the position and settings were determined, a counterweight was locked into place to neutralize the gravitational forces of the upper body (head, torso, arm). Furthermore, patients were tested for limitations in their lumbar range of motion (ROM) between 0° and 72°. Since studies have shown that ILEX with limited ROM is equally effective as compared to full ROM in increasing extension strength and can nonetheless result in perceived pain reductions [25], the exercise ROM was adjusted to patients’ mobility and symptoms (Figure 2). To reduce mechanical stress on lumbar discs and/or the nerve root, patients in acute (inflammatory) stages were more distinctly restricted in flexion in this study (average 42°) [22].

After the apparatus had been adjusted to the patients’ anthropometric characteristics, an isometric extension strength test was performed for most patients. Patients that had extremely severe pain and were in a very acute stage were not tested for safety reasons. In order to determine the isometric extension strength, four positions within the chosen ROM were tested. Patients were instructed to extend their back against the upper back pad by gradually building tension over a two to three second period. Once maximal tension was achieved, the contraction was maintained for one second before relaxing. Between each isometric contraction, a rest period of about 10 s was provided while the patient was moved softly between flexion and extension several times. The testing procedure was repeated at the end of the rehabilitation program.

### 2.4. Exercise Protocol

In general, the training consisted of a dynamic lumbar extension resistance exercise with high intensity and low frequencies. Studies on the efficiency of ILEX have revealed that one to two exercise sessions per week and one set per session with high intensities targeting momentary muscular failure provide sufficient training stimulus to restore lumbar extension strength [26]. In this study, 18 sessions (two sessions per week) were applied. Each flexion–extension cycle during ILEX training lasted for around 10 s (4 s extension, 2 s holding in maximal extension, 3 s flexion) providing for a safe, controlled movement and a sustained time under tension of the muscles. Within the first six sessions, patients were trained with submaximal loads gradually adapting to the training regimen. Sessions one to nine were characterized by continuous increase of both resistance load and ROM. From session 10 onwards, the exercise load was set high enough to achieve total muscle fatigue after 12–15 repetitions. For optimal muscle control and in order to avoid fast, uncontrolled movements potentially harming the patient, the speed of each flexion–extension cycle was guided by a benchmark on the screen. The increase of training weights and the modifications of the ROM were set in accordance with the patients’ current pain status and wellbeing. In addition to the ILEX training, patients performed four different exercises (one set, 12–15 repetitions) to improve core stabilization and to condition superficial back muscles. These exercises were customized to the individual situation of the patients. The most common exercises were abdominal crunch (only in non-acute patients!), horizontal rowing, standing cable pull, reverse butterfly and latissimus pull-down. Patients were permanently supervised by one experienced therapist.

### 2.5. Outcome Measures

Before the first and after the last training sessions, questionnaires were filled out by the patients. The current pain/radiculopathy (SR) status was rated by a numeric rating sale (0–10, 0 = no pain, 10 severe pain). Using the same scale, impact on mental health (IOMH) (0 = no impact, 10 = severe impact) and satisfaction with the clinical results (0 = not satisfied, 10 very satisfied) was measured. The overall progress of the rehabilitation (clinical outcome) was stated in percentage of (0% = no change, no improvement; 100% = complete relief of symptoms). Furthermore, patients were asked to judge on the clinical outcome by one of the following terms: free of symptoms, marked improvement, light improvement, no change, worsening of symptoms.

### 2.6. Statistics

SPSS was used to analyze the data (SPSS, Chicago, IL, USA). Normal distribution was tested using the Kolmogorov–Smirnov test. Non-parametric tests were used to analyze clinical outcome (Wilcoxon test) and to determine differences in clinical results between groups (Mann–Whitney U, Kruskal–Wallis with Bonferroni correction). Spearman correlation coefficient was calculated for determining whether pain scores, age or achieved strength increase correlated with clinical outcome and satisfaction. Furthermore, a multiple linear regression was calculated to predict clinical outcome for these factors. Significance level was set to *p*-value < 0.05

### 2.7. Ethical Consideration

The study was approved by the local ethics committee (Sports Science Institute, Würzburg University, Würzburg, Germany). All patients signed an informed consent. 

## 3. Results

After 18 sessions, 162 out of 168 patients (96.4%) reported significant reduction of clinical symptoms: 146 patients (86.9%) were completely free of symptoms or had some remaining minor symptoms. No patients had a worsening of symptoms (Figure 3). 

Pain and radiculopathy symptoms (NRS) decreased on average from 4.2 (±1.5) to 1.9 (±1.5) (*p* < 0.001) (−54.8% reduction), and the impact on mental health (suffering) decreased from 5.9 (±2.3) to 2.4 (±2.0) (*p* < 0.001) (−59.4% reduction) (Figure 4a). On NRS scales, the impact on mental health was rated significantly higher than the physical symptoms (Figure 4b).

The satisfaction rate was, on average, 8.5 ± 2.0 (range 0–10), and the clinical outcome was rated 78.9% ± 20.4 (range 0–100). The precise distribution of these two parameters is shown in Figure 5a,b. Isometric lumbar extension strength improved by 35.7% on average.

There was a weak correlation between lower levels of pain and mental suffering prior to the treatment, and better clinical results in terms of rehabilitation progress. Lower pain levels before the program also correlated weakly with the satisfaction of the treatment (Table 1). There were no correlations with the parameters age and strength increase. Multiple linear regression analysis showed that there was significant regression for clinical outcome (in %) (F(4118) = 3.902, *p* < 0.01), with an R^2^ of 0.117. Clinical outcome decreased by 3.40% for each point higher on the NRS scale for pain and radiculopathy (PR), which was therefore a weak but significant predictor (*p* < 0.01) of clinical outcome. Similar results were found for satisfaction (F(4118) = 3.174, *p* < 0.05, R^2^ = 0.097). For each point higher on NRS for PR, the satisfaction decreased by 0.33 points (*p* < 0.05). There was no significant prediction for the other factors IOMH, age and strength increase.

The affected disk level and the duration of symptoms had no impact on the results (Table 2). There was a trend for acute patients (<3 months of symptoms) showing better outcome than in chronic patients.

Patients that dropped out due to persisting high pain levels (*n* = 21) had significantly higher levels of PR and IOMH than those who finished the therapy program (Figure 6).

## 4. Discussion

This is, to our knowledge, the first study that has examined the clinical outcome of an ILEX-based exercise therapy program on a large series of patients with properly diagnosed lumbar disk herniation on MRI exhibiting radiculopathy and sensory impairment. The results of the study showed that almost all patients (162/168) who finished the 9-week program were safely and successfully treated with this ILEX-based exercise program. None of the patients who finished the program reported a worsening of symptoms. There was a significant reduction of average scores related to pain and radiculopathy symptoms and influence on mental health. Furthermore, there were high scores for clinical outcome and satisfaction rate. Among the analyzed factors, only high levels of pain and impact on mental health showed a (weak) negative correlation with the clinical outcome. For pain, there was also a weak correlation with satisfaction with the treatment. Furthermore, pre-interventional pain levels were a predictor for the clinical outcome and satisfaction rate based on linear regression analysis. Patients who dropped out during the process due to medical reasons had significant higher PR and IOMH scores prior to therapy start than those from the final case series.

There is still a debate whether surgical or non-operative treatment is best for patients with LDH, especially in the long-term perspective. In recent meta-analyses, some low evidence (considered as low quality) indicated surgery being more effective than prolonged non-operative treatment, considering short-term pain relief and improved physical function [27,28]. However, studies evaluating long-term outcomes have shown no significant differences between the two treatments after one, two and five years [9,10,28]. The authors concluded that prolonged conservative management might give patients a good chance to become pain-free without surgery, but some authors also highlight the risks of delayed surgery after prolonged suffering from sciatica. Across all these studies, it becomes clear that there is a lack of high-quality interventions and that there is a variety of different treatment approaches, making it difficult to rate the effectiveness of conservative treatments in general. Therefore, clinicians are faced with the difficulty of choosing the right treatment option for patients with LDH. Furthermore, there is a lack of research on customized treatment approaches. Some of ILEX main advantages are that it can standardize ROM and can constantly be adapted to patients’ complaints, which allows a pain-free treatment even in acute stages.

Looking at the existing studies on exercise interventions in LDH patients, the majority has applied different types of core stabilization and motor control exercises. The results indicate positive effects on lumbar disk function restoration and pain reduction [18,19,20,21,22,29]. Furthermore, in an RCT by Ye and colleagues, lumbar spine stabilization exercises have shown to have superior effects compared to general exercise [18]. Despite these promising results for exercise interventions, there are also some major difficulties. First of all, there is only a limited number of comprehensive studies to date; second, comparative evaluation of treatments proves to be difficult due to different standardization forms; finally, none of the conservative treatment options has this far shown continuous success throughout different patient subgroups and qualifies as gold standard [8]. According to Hahne and colleagues, this discrepancy and variety in treatment responses between individuals may be due to a high heterogeneity among patients. Since many studies have diagnosed LDH without using an MRI by neurological examinations alone, they have often failed to exclude patients with other potential pain sources such as degenerative stenosis, spondylolisthesis and (age-related) facet joint arthritis [8]. To avoid this drawback, all patients included in this study were neurologically examined by an experienced medical consultant and spine surgeon and also presented an MRI that was not older than 3 months. Further, patients were not older than 50 years of age and had no additional pathological changes in the spine on MRI. With this study approach, we were able to establish a homogenous patient population guaranteeing a proper examination of the interventional effect on LDH.

The pathophysiological mechanisms of development and spontaneous recovery of LDH are not fully understood. However, it is well established that the probability of spontaneous regression is associated with the type of herniation. The rate and occurrence of regression has been found to be 96% for disk sequestration and 70% for disk extrusion. The rate of complete resolution of disk herniation has been shown to be 43% for sequestrated disks and 15% for extruded disks [30]. According to recent investigations, it is likely that an inflammatory response against the free fragments is one key mechanism and a good prognostic indicator for the regression process including matrix remodeling and neovascularization [31,32]. Zhong and colleagues compared the rate of spontaneous regression (overall mean 67%) between studies from different countries and concluded that Western countries with potentially advanced medical systems and standardized conservative treatment algorithms, such as the UK (mean 83%), have higher rates of disk regression [11]. Thus, the type of conservative treatment method has most likely significant effects on the regression of LDH and on the clinical improvement of patients by increasing spinal stability. In any case, there seems to be a good chance for LDH patients to recover from their symptoms without surgery. Therefore, conservative measures should be first-line treatment. Early surgery may be required when patients exhibit significant impairment of motor functions and/or bladder and bowl control. Another exception may be patients suffering from lateral disk herniations [33]. For this quite rare condition—which accounts only for 7–12% of LDH patients—conservative management is often not successful according to Epstein [33]. The six patients in our study who did not benefit from ILEX training had medial or medial-lateral disk herniations. Reasons why these patients did not report a reduction in pain symptoms remain unclear.

It has become evident that chronic pain conditions in the lower back are associated with a deconditioning process in the paraspinal back muscles and that selective strengthening of these muscles should be one major target in the rehabilitation process for this patient group [12]. Indeed, excellent clinical outcome has been demonstrated in multiple studies [14]. When comparing different exercise approaches, there is some evidence that the best isolation of the paraspinal muscle can be achieved by appropriate pelvic restraint [15]. According to Steele and colleagues, the system used in the current study can provoke higher muscle fatigue than other approaches [15]. In a study by Fortin et al., the relationship between muscle atrophy and local fat infiltration in the paraspinal muscle, and the level and location of LDH and symptoms was shown [16]. Thus, there is a rational for applying ILEX for the rehabilitation of LDH patients too. Beside the isolation of paraspinal muscles, another advantage of ILEX training compared to other exercise types is that it can be standardized by adjusting the ROM (e.g., reducing movement in flexion) and resistance load to ensure safety for the patient, especially in the acute stages. This adaption is based on the findings from cadaver studies highlighting that the amount of compressive force and tension in the nerve root increases with flexion of the spine [34]. Yet, the physiological mechanisms behind the clinical effects of exercise interventions are not completely understood. Based on the anatomical situation of paraspinal back muscles (e.g., multifidus) and their function as spine stabilizer, it is very likely that regaining muscle strength enhances spinal stability, which may facilitate the healing process of the disk. This aspect needs to be explored in further studies. As radiographic resolution of LDH is expected to be seen on average after 9–10 months [32], it is also necessary to follow up patients via MRI. Based on the current data, it cannot be ruled out that some patients would have been cured by spontaneous LDH remission through wait and see. However, as all patients had tried other treatments without success, it is likely that ILEX was needed to stabilize the spine, which may enhance the probability of regression of LDH.

The results of this study clearly show that ILEX with adapted ROM and heavy resistance load is a very promising tool for the rehabilitation of patients with LDH and radiculopathy. Due to its standardizing character, it may also serve as a safe postoperative rehabilitation measure to regain strength and prevent from recurrent LDH. For appropriate decision making, it is important to note that higher pre-interventional scores for PR and IOMH weakly correlate with lower rehabilitation progress. In our initial material, the 21 patients that dropped out due to persistent pain and/or surgery had on average significantly higher scores for pain and impact on mental health compared to those that finished the program. Thus, for patients with extremely high pain levels, clinicians should keep in mind that surgery is perhaps needed, and the right choice should be evaluated after each exercise session for those patients. For all other patients, it is very likely that ILEX can successfully relief patients’ symptom. Surprisingly, the disk level of LDH and age had no impact on the results. Another observation (not shown in the study) was that also the size and location (medial vs. mediolateral) of LDH did not seem to have an impact. However, there was a non-significant trend for acute patients (<3 months of symptoms) having better results. Thus, despite the promising results of ILEX for almost all LDH patients, we believe that treatment should ideally start early during the process of pathogenesis.

One limitation of the current study’s design is that several patients (*n* = 21) did not finish the therapy and were excluded from the final analysis. For some patients (*n* = 14), the decision to interrupt the therapy and instead undergo surgery was made by the local consultant, while for the other patients that stopped the treatment (*n* = 7), there was no clinical indication. However, the inclusion of these dropouts can also be seen as a strength of this study, as it reflects the real situation in clinical practice. It shows that not all patients can recover and avoid surgery, but those who are able to stick to the process of ILEX-based rehabilitation have high chances of good clinical results. Another limitation is the impact of painkillers (in 31% of patients) that has likely influenced the patients’ pre-interventional pain scores and isometric strength tests. For some patients, it was no option to stay away from medication, especially in the beginning of the therapy. However, during the program patients could gradually decrease the dose. Another limitation is that additional conventional exercises were added to the therapy program. Thus, the clinical outcome cannot be ascribed solely to ILEX. Furthermore, it can possibly be criticized that patients beyond 50 years of age were excluded. This setpoint for age was determined by expert consultation. We hypothesized that patients beyond 50 years of age would exhibit increasing signs of spine degeneration that would likely influence pain perception. Finally, this study is a case series and lacks comparison to a control group and/or another treatment type. Randomized controlled studies comparing the clinical outcome of ILEX in limited ROM to other surgical and conservative treatments for patients with LDH and radiculopathy are warranted. Further research should also focus on gaining new insights that may help to explain the mechanisms responsible for positive clinical outcomes.

Despite these limitations, the study can deliver important insights on the outcome of ILEX and the benefits of exercise in general on herniated disks, as it reflects the real situation in many clinical practices. All patients had tried other conservative methods and were referred to the spine center for potential surgical intervention. The results can help clinicians making appropriate decisions and highlight a new treatment approach that is safe and promising. Further studies using imaging for monitoring rehabilitation progress are now needed to shed more light on the mechanisms behind the good clinical outcome. Furthermore, the potential as a post-interventional program after surgery or injection treatment needs to be evaluated.

Although not specifically addressed in this study, there is a high chance that aspects relevant in the context of the psychosocial nature of back pain are at least partly responsible for the clinical outcome. As such, it is possible that the controlled movement in pain-free ROM during ILEX changes dysfunctional cognitions modulated by fear of pain (e.g., kinesiophobia, fear-avoidance behaviour). Thus, the patient can reconsider his attitudes and behavior, thereby regaining trust in the capabilities of his body. In this context, the role of the therapist needs to be highlighted, as he is responsible for creating an atmosphere of trust and safety for the course of the whole therapy. Comparison of pre/post scores show that the reduction of IOMH was larger than that of PR, which may be an indicator for an increased trust in patients’ own body but also in the therapy as such.

## 5. Conclusions

In conclusion, the current study is the first to show evidence and safety for the effective treatment of patients with LDH and radiculopathy using ILEX. Among the analyzed factors, only high pre-interventional pain scores and impact on mental health showed a weak negative correlation with the clinical outcome. Due to the possibility of standardization of ROM, it may also serve for other patient groups with neuropathic pain conditions and for post-operative rehabilitation. The results can support orthopedic consultants in decision making.

## Figures and Tables

**Figure 1 jcm-10-02430-f001:**
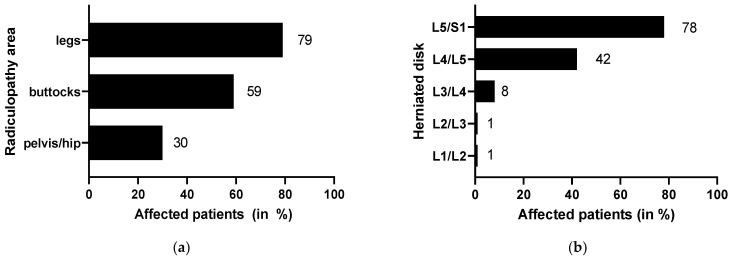
Characteristics of symptoms related to the area of radiculopathy (**a**) and the affected disk level (**b**).

**Figure 2 jcm-10-02430-f002:**
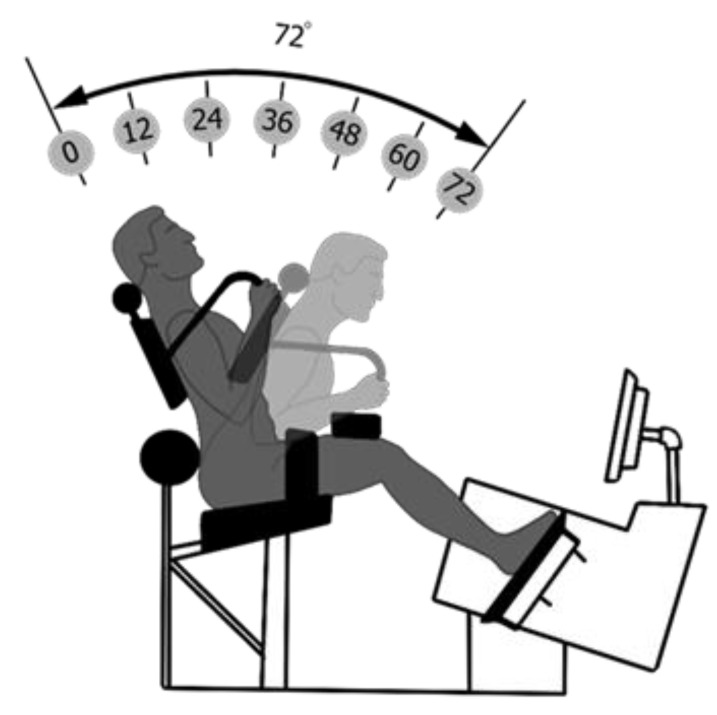
Exercise machine and its restraint system. Training in limited range of motion following a benchmark on the screen.

**Figure 3 jcm-10-02430-f003:**
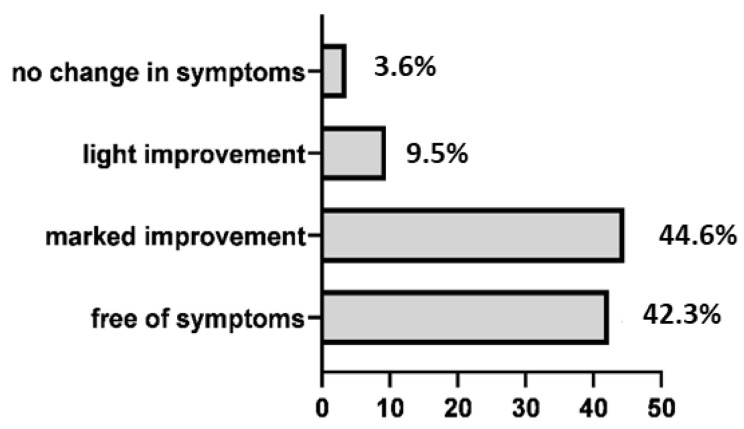
Overview of subjective evaluation of clinical results (in %).

**Figure 4 jcm-10-02430-f004:**
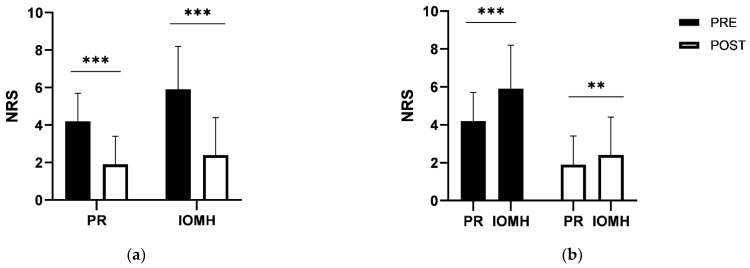
Change in symptoms (pain/radiculopathy scores, PR) and impact on mental health (IOMH) before and after the rehabilitation program (**a**). Comparison between PR and IOMH before and after the study (**b**). (** *p* < 0.01, *** *p* < 0.001).

**Figure 5 jcm-10-02430-f005:**
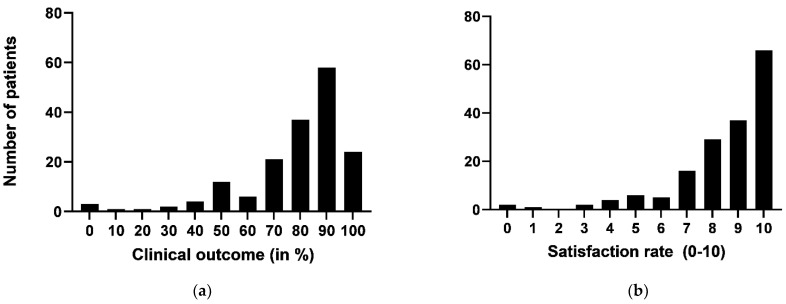
Clinical outcome (**a**) and satisfaction rates (**b**) related to the therapy program (in number of patients).

**Figure 6 jcm-10-02430-f006:**
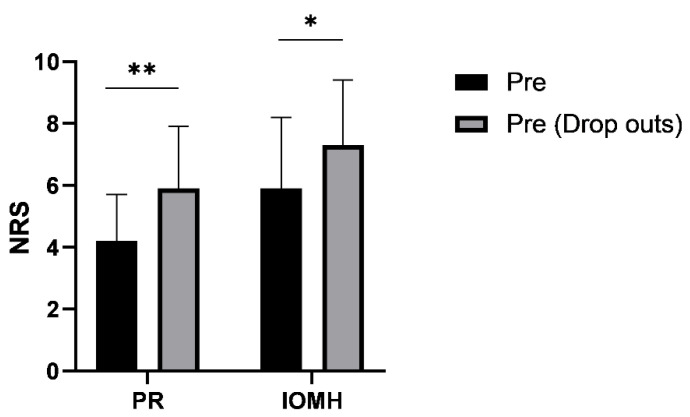
Comparison of PR and IOMH (pre, before the program) between patients who finished the program and those who dropped out. (* *p* < 0.05, ** *p* < 0.01).

**Table 1 jcm-10-02430-t001:** Analysis of Spearman correlation coefficient between pain (pre), perceived impact on mental health (pre), age and strength increase with overall clinical outcome and satisfaction rates. (** *p* < 0.01, *** *p* < 0.001).

	Clinical Outcome (in %) (Spearman Rho)	Satisfaction Rats (Spearman Rho)
pain (pre)	−0.261 ***	−0.207 **
impact on mental health (pre)	−0.207 **	−0.139
Age	−0.034	−0.020
strength increase	0.098	0.041

**Table 2 jcm-10-02430-t002:** Comparison of outcome measures (mean ± standard deviation) in patients with different affected disc levels and different durations of symptoms.

	Clinical Outcome (in %)	Satisfaction Rats
**Affected disk**	*p* = 0.771 ^a^	*p* = 0.822 ^a^
L5/S1	78.60 ± 19.14	8.51 ± 1.77
L4/L5	80.97 ± 13.50	8.74 ± 1.48
L5/S1 + L4/L5	78.00 ±23.84	8.43 ± 2.22
**Duration of symptoms**	*p* = 0.282 ^a^	*p* = 0.317 ^a^
<3 months	82.24 ± 15.65	8.88 ± 1.23
3–12 months	75.91 ± 21.17	8.23 ± 2.27
>12 months	75.96 ± 24.04	8.12 ± 2.34

^a^ Kruskal Wallis test.

## Data Availability

The datasets generated and analyzed during the current study are not publicly accessible but are available from the corresponding author on reasonable request.

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
