# Peer review of "Isolated Lumbar Extension Resistance Exercise in Limited Range of Motion for Patients with Lumbar Radiculopathy and Disk Herniation—Clinical Outcome and Influencing Factors"

_jcm, 2021, doi:10.3390/jcm10112430_

Round 1

Reviewer 1 Report

very interserting and well written article

in discussion i am missing a few things 

1: what could be the impact of spontaneous remission during the periode of weeks the exercizes take place ??

2: i am missing future direction

a: an rct is needed comparinh ILEX to conservative management

b: what could be the extra value of ILEX after a trasforaminal epidural ( would live to run these studies ) 

Reviewer 2 Report

An interesting topic within the complicated area of lumbar radiculopathy and disk herniation.

The title: Isolated lumbar extension resistance exercise in limited range of motion for patients with lumbar radiculopathy and disk herniation – A case study on clinical outcome and influencing factors  The yellow marked words can be taken away it will then be; Isolated lumbar extension resistance exercise in limited range of motion for patients with lumbar radiculopathy and disk herniation – Clinical outcome and influencing factors.

In Figure 1b missing percent figures for L5/S1, L4/L5

Table 1 is confusing, also together with the text above line 250..., with use of the Pearson correlation coefficient and then a p-value for age and strength? Why not use Spearman's test instead of Pearson? This part must also be better clarified, no text about this under "Statistics". The significance doesn't add anything, can be taken away. I agree that the correlation is weak, but from which reference do they lean against? 

Line 312 Hahne it should be Hahne and collegues

Line 319 ag it should be age

Round 2

Reviewer 2 Report

Line 240 it says ...the impact on mental health (suffering) decreased from 5.9 (±2.3) to 2.4 (±2.0) (p<0.001). In the abstract it says, line 29/30......the impact on mental health decreased from 5.5 (±2.4) to 2.3 (±1.7) (p<0.001)

In the abstract line 30-33 There was a correlation between lower scores for PR and IOMH before the study and better clinical outcomes and satisfaction. Other factors such as age, strength increase, level/location and number of LDH did not have a significant impact on clinical results. Is this in line with the results on page 7 line 54-65? in relation to weak and in terms of rehabilitation progress. Suggestion to go through the abstract and the discussion in relation to the text.

I don't think it is necessary with the significance in the Table 1 the important figures are the correlation and it is that you report?

In Figure 1b figures are added in one out of two, still missing percent figures for L5/S1, L4/L5.

Author Response

Line 240 it says ...the impact on mental health (suffering) decreased from 5.9 (±2.3) to 2.4 (±2.0) (p<0.001). In the abstract it says, line 29/30......the impact on mental health decreased from 5.5 (±2.4) to 2.3 (±1.7) (p<0.001)

Thank you. The numbers in the abstract were indeed not correct. We went through our calculations and also through the whole manuscript carefully and have adjusted those numbers.

In the abstract line 30-33 There was a correlation between lower scores for PR and IOMH before the study and better clinical outcomes and satisfaction. Other factors such as age, strength increase, level/location and number of LDH did not have a significant impact on clinical results. Is this in line with the results on page 7 line 54-65? in relation to weak and in terms of rehabilitation progress. Suggestion to go through the abstract and the discussion in relation to the text.

Thank you. Again, the abstract was by mistake not updated and correct. We have changed it now. Due to the word limit in the abstract we have only included the correlation aspect in the abstract and not the regression analysis.

I don't think it is necessary with the significance in the Table 1 the important figures are the correlation and it is that you report?

We have read in the literature and the p values are important as the indicate if the correlation is significantly different from 0. As our values are quite small we believe that it is necessary to keep the significant levels.

In Figure 1b figures are added in one out of two, still missing percent figures for L5/S1, L4/L5.

We have replaced the figures. In the „marked“ version it may have been a little confusing. In the latest „clean“ version these values should be included.